# Convergent and Concurrent Validity between Clinical Recovery and Personal-Civic Recovery in Mental Health [note 1]

**DOI:** 10.3390/jpm10040163

**Published:** 2020-10-12

**Authors:** Jean-François Pelletier, Larry Davidson, Charles-Édouard Giguère, Nicolas Franck, Jonathan Bordet, Michael Rowe

**Affiliations:** 1Centre de Recherche de l’Institut Universitaire en Santé Mentale de Montréal, 7401 Hochelaga Street, Montreal, QC H1N 3M5, Canada; cedouard-giguere.iusmm@ssss.gouv.qc.ca (C.-É.G.); jonathan.bordet.cemtl@ssss.gouv.qc.ca (J.B.); 2Yale Program for Recovery & Community Health, Erector Square, Building 1, 319 Peck Street, New Haven, CT 06513, USA; larry.davidson@yale.edu (L.D.); michael.rowe@yale.edu (M.R.); 3Université Claude-Bernard Lyon 1, 43 Boulevard du 11 Novembre 1918, 69100 Villeurbanne, France; nicolas.franck@ch-le-vinatier.fr; 4Centre hospitalier Le Vinatier, 95 Boulevard Pinel, 69678 Bron, France

**Keywords:** patient-reported outcome measures, patient-developed outcoworkersme measures, clinical recovery, personal recovery, civic participation, peer support

## Abstract

Several instruments have been developed by clinicians and academics to assess clinical recovery. Based on their life narratives, measurement tools have also been developed and validated through participatory research programs by persons living with mental health problems or illnesses to assess personal recovery. The main objective of this project is to explore possible correlations between clinical recovery, personal recovery, and citizenship by using patient-reported outcome measures. All study participants are currently being treated and monitored after having been diagnosed either with (a) psychotic disorders or (b) anxiety and mood disorders. They have completed questionnaires for clinical evaluation purposes (clinical recovery) will further complete the Recovery Assessment Scale and Citizenship Measure (personal-civic recovery composite index). Descriptive and statistical analyses will be performed to determine internal consistency for each of the subscales, and assess convergent-concurrent validity between clinical recovery, citizenship and personal recovery. Recovery-oriented mental health care and services are particularly recognizable by the presence of Peer Support Workers, who are persons with lived experience of recovery. Upon training, they can personify personalized mental health care and services, that is to say services that are centered on the person’s recovery project and not only on their symptoms. Data from our overall research strategy will lay the ground for the evaluation of the effects of the intervention of Peer Support Workers on clinical recovery, citizenship and personal recovery.

## 1. Introduction

Recovery is now the official leading paradigm in the transformation of mental health systems and policies in the UK [1], the USA [2], Canada [3], and elsewhere around the world [4]. For 15 years, recovery has also been at the core of successive Mental Health Action Plans in the province of Quebec, Canada [5,6]. As a social movement echoing the historical claims of other social movements since the 1960s and 1970s, including the antipsychiatry movement, the origins of recovery in mental health are now fairly well documented [7,8]. Nevertheless, tensions persist about the meaning and ownership of recovery [9,10].

Generally speaking, there are two major portrayals of recovery [11,12]. One is akin to the notion of cure in the field of physical health; (A) clinical recovery indeed refers primarily to the reduction of psychiatric symptoms through a curative approach to the disease using psychopharmacology and psychotherapy. With this first axiom of recovery, the role of the ill person is mainly to follow the instructions of professionals and comply with prescribed treatments. On the other hand; (B) a more personal axiom of recovery promotes the empowerment of the persons, their ownership and authorship of their own history, autonomy, and independence in living free from any labeling diagnosis [13,14]. Here, living with the condition is seen as a continuous learning opportunity through which a person can profoundly transform himself of herself, even to the point of not wanting to be cured in the sense of returning to the same state as before the onset of that condition. This is especially true when this condition is associated with harmful lifestyles (e.g., abusive substance use and subsequent depression).

Recovery is not the absence of symptoms but a redefinition of oneself in light of lived experience as a person living with mental health problems or illnesses (MHPIs) who found a new balance in life towards wellbeing, with or without psychiatric diagnosis, medication or treatment. Although they are often presumed to converge, recovery from mental illness and recovery in mental health need to be distinguished. Several instruments have been developed by clinicians and academics to assess clinical recovery. Based on their life narratives, measurement tools have also been developed and validated through participatory research programs by persons living with MHPIs to assess citizenship and personal recovery, namely, the Recovery Assessment Scale [15] and the Citizenship Measure [16] questionnaires.

Cross-sectional studies have been conducted [17] to analyze data from persons living with MHPIs but at a specific point in time (including with the Recovery Assessment Scale). To evaluate the outcomes of recovery-oriented interventions on clinical and personal recovery combined to citizenship, a validated measure of organizational recovery is also needed. The Recovery Self-Assessment was thus designed to gauge the degree to which programs implement recovery-oriented practices [18]. It is a self-reflective tool designed to identify strengths and target areas of improvement as agencies and systems strive to offer recovery-oriented care. The RSA contains concrete, operational items to help program staff, persons in recovery, and significant others to identify practices in their mental health and addiction agency that facilitate or impede recovery.

In the case of enduring MHPIs, the information offered to persons diagnosed with psychiatric conditions need not be limited to the nature or etiology of the underlying pathology but should include information about how to live as satisfactorily and as independently a life as possible in spite of the persistence of these conditions while continuing to strive to reach one’s full citizenship (citizenship is to be distinguished from nationality). Here the person living with MHPIs is considered an end-user in the knowledge translation circuit. Combined with accurate health information, the experience of living in recovery in the long term in the community is particularly useful for sharing among peers who are coping, and/or have coped, with similar issues. The commonality is the struggle and emotional pain that can accompany the feeling of loss and/or hopelessness due to MHPIs, rather than in relation to a specific symptom or illness. Peer-to-peer communication is a widespread phenomenon, for example in groups like Alcoholics Anonymous. Then, Peer Support Workers (PSWs) are persons who are further along in their own recovery journey. Upon training, they can provide supportive services, for instance when hired to fill such a paid specialty position directly in, or in conjunction with, mental health service provision [19]. PSWs provide the mental health service users–their mentees–a validation of their lived experience and experiential knowledge for facilitating the reclaiming of their lives in the community [20,21]. The Mental Health Commission of Canada suggests that patients living with MHPIs who interact with PSWs will not only feel the empathy and connectedness that comes from similar life experiences, but that this interaction also fosters hope in the possibility of a recovery that includes health, wellbeing, quality of life, and resilience [22]. Indeed, patients served by case management teams with PSWs have shown greater treatment engagement, more satisfaction with their life situation and finances, and fewer life problems than in comparison to case management alone [23,24]. However, it is mainly the effect of the intervention of teams or programs where there are PSWs that has been evaluated so far. It remains difficult to isolate the PSWs’ intervention in order to attribute a specific effect of this intervention in terms of clinical recovery, citizenship and personal recovery on patients living with MHPIs. The main aim of this study is thus to firstly explore convergent and concurrent validity between these constructs.

## 2. Materials and Methods

### 2.1. Study Design and Population

The protocol of this study was approved by the Ethics Committee of Institut universitaire en santé mentale de Montréal (IUSMM) (protocol # 2020-1948). Indeed, we will recruit our study participants, whether living with (a) psychotic disorders or (b) anxiety and mood disorders, from among those who have already agreed to participate in the IUSMM Signature Bank. The “signatures” of MHPIs is a term formulated by the National Institute of Mental Health to designate the broad range of genetic, biological, psychological, and social factors that may “sign” a specific mental disorder, depending on an individual’s sex, history, lifestyle habits, and so on [25]. In 2010, based on the recommendations of an international advisory committee composed of some of the best scientists in the world in the field of psychiatric research, the Research Center of IUSMM decided to develop the “Signature Bank” project for the collection of biological and dimensional signatures from all patients with MHPIs of the IUSMM (catchment area of about 600,000 inhabitants). Over 4000 patients are treated annually at the IUSMM, while an additional 2000 patients per year are treated by means of outpatient or ambulatory services. This is one of the largest populations of psychiatric patients in Canada. What is unique about this ambitious research project is the extensive involvement of the IUSMM-hospital site in the attempt to establish an exclusive niche for discoveries in the signatures of mental illnesses. By collaborating with the Research Centre, IUSMM-hospital managers have contributed to the implementation of this large-scale project that aims at measuring the (epi)genetic, biological, psychological, and social signatures of people living with MHPIs who receive the IUSMM-hospital’s clinical services, and who consent to taking part in this longitudinal research initiative that brings together clinicians, patients living with MHPIs, and researchers. This proposal research project goes even further in understanding not only the signatures of MHPIs, but recovery in mental health, and as reported by patients who will additionally fill out (i) the Recovery Assessment Scale, (ii) the Citizenship Measure and (iii) the Recovery Self-Assessment.

#### 2.1.1. Sample Selection and Procedure

The research objectives will be achieved with participants of the Signature Bank who have already accepted to be contacted for such purposes and therefore, who have already completed measures iv-ix described below [26]. Participants in the Signature Bank are approached and recruited by a Research Nurse at their admission to the Psychiatric Emergency Department (T1) of the IUSMM. As of March 2019, 1862 eligible participants from the psychiatric emergency of the IUSMM have been approached. Of this number, 1218 agreed to participate and thus completed at least T1. For all participants, the same iv-ix measures are repeated at discharge (T2), at the first follow up at an outpatient clinic (T3), and finally (T4) when treatment ends or 12 months after T3. Our sample is characterized by individuals with psychotic disorders (N = 166) or mood disorders (N = 186) for a total of 352 eligible patients who completed T3 and/or T4. All participants signed a detailed consent form, and the study was approved by the local ethics committee in accordance with the Declaration of Helsinki. Research nurses collected patient’s psychiatric diagnoses from medical records. They were established by psychiatrists on the ward and coded according to the World Health Organization International Classification of Disease (ICD-10) [27]. This study will use 2 of the categories of mental or behavioral disorder (categories F00-F99 of ICD10): (1) Schizophrenia and psychotic disorders (categories F20-F29), and (2) Anxiety or mood disorders (categories F30-F49). It is possible for any of the users of IUSMM clinical services (including the hospital’s outpatient clinics) to participate in the Signature Bank data collection. Each patient diagnosed with F20-F49 disorders will be contacted by phone and asked to additionally fill out the RSA, RAS, and CM (measures i–iii). Those who will accept to be contacted will be invited to come to the IUSMM where they will be met by a Research Assistant. They will first read and sign the additional Information and Consent Form specific to this study or ask further questions before doing so. Secondly, they will fill out measure i–iii on a touch screen device. Data will be stored on the Signature Bank secured server. Thirdly, they will receive $20 as compensation for their time.

#### 2.1.2. Exclusion Criteria

Exclusion criteria will be: patients with active suicidal thoughts.

#### 2.1.3. Sample Size Estimation

A sample-size determination analysis was done in G*power v. 3.1.9.4. In the convergent analyses, correlation should be superior to 0.3 in absolute value. To detect an effect of this magnitude or greater, using a 5% type I error, we need at least 82 participants. Increasing the sample by 15%, to be conservative, leads to a sample-size of 95 patients to perform the planned analyses [28,29].

### 2.2. Measurements

Using a translation–back-translation method [30], our team has translated from English to French three patient-reported outcome measures, respectively, (i) the Recovery Assessment Scale (RAS), (ii) the Citizenship Measure (CM), and (iii) the Recovery Self-Assessment (RSA) [31]. Convergent and discriminant validity are both subtypes of construct validity. Given that the CM and the RAS are measures of constructs that theoretically should be related, convergent validity has been tested and already confirmed between the CM and the RAS by estimating correlation coefficients. It was concluded that the CM demonstrates convergent validity with the RAS, and we suggested combining them within the concept of civic recovery [32]. Indeed, the psychometrics of the new French-versions of the RAS and CM have been evaluated among French speaking research participants in the province of Quebec (n = 174). The internal consistency of each scale, from the exploratory factor analysis (new CM) and from the confirmatory factor analysis (well established RAS), respectively, was assessed using Cronbach’s alpha. Pearson correlations were calculated between the dimensions to assess the tools’ convergent validity (Table 1).

We now plan to estimate correlation coefficients between the CM and RAS on one side (total of 47 items for this personal-civic recovery composite index), with six measures of clinical recovery on the other side (total of 59 items). In total, 95 participants were met between 1 September 2019 and March 1 2020. They all completed the abovementioned questionnaires and quantitative analyzes are on the way. We expect to submit the main findings by the end of the year 2020.

#### 2.2.1. Recovery Assessment Scale

The RAS is a 24-item questionnaire with 5-point Likert scales. Higher scores are positively correlated with higher levels of recovery. Minimum score = 24; maximum score = 120. Salzer and Brusilovskiy have published an in-depth review of the quantitative properties of the RAS, based on 77 articles that included psychometric data. They concluded that these studies indicate very good results for internal consistency, test-retest reliability, and internal reliability [33]. Among the tools available to empirically assess recovery, the RAS has been the most published. The RAS items spread over the following five dimension scales: (1) Personal confidence (9 items, Cronbach’s alphas = 0.86), (2) willingness to ask for help (3 items, Cronbach’s alphas = 0.83), (3) goal and success orientation (5 items, Cronbach’s alphas = 0.68), (4) reliance on others (4 items, Cronbach’s alphas = 0.65), and (5) no domination by symptoms (3 items, Cronbach’s alphas = 0.73).

#### 2.2.2. The Citizenship Measure

The CM is a 23-item questionnaire (short version) with 5-point Likert scales. Higher scores are positively correlated with higher levels of citizenship. Minimum score = 23; maximum score = 115. It was developed through a Community-based Participatory Research design in response to a prompt, suggested by persons living with MHPIs who were involved as research partners and research staff. The prompt was “For me, being a citizen means…” The CM items spread out on the following five dimensions: (1) Self-determination (6 items, Cronbach’s alpha = 0.67), (2) respect by others (4 items, Cronbach’s alpha  =  0.74), (3) involvement in community (4 items, Cronbach’s alphas = 0.65), (4) basic needs (5 items, Cronbach’s alpha  =  0.60), and (5) access to services (4 items, Cronbach’s alpha  =  0.60).

#### 2.2.3. The Recovery Self-Assessment

The RSA was designed to assess their current experience of mental healthcare. This tool was developed to gauge the degree to which programs implement recovery-oriented practices and to identify strengths and areas of improvement as agencies strive to offer recovery-oriented care. This is a 32-item questionnaire with 5-point Likert scales. Higher scores are positively correlated with higher levels of recovery-oriented care. Minimum score = 32; maximum score = 160. The RSA is among the most widely used rating scales to facilitate reflection on the strengths and limitations of services within a recovery framework [34]. It is intended for use with individuals who receive and/or provide services in inpatient settings, outpatient settings, peer-run programs, residential programs, and social programs. The RSA questionnaire has versions for administrators, service providers, family members/key supports, and consumer. It is this latter consumer version that we will use in our own study. RSA items cover five domains: Life goals versus symptom management; Consumer involvement and recovery education; Diversity of treatment options; Rights and respect; and Individually tailored services. The RSA allows for a generation of a total mean score, domain means, and for the comparison of stakeholder perspectives [35,36]. The consumer version of the RSA assesses the perceptions of individuals with lived experience about whether the system and its providers embrace the core principles of recovery.

#### 2.2.4. Measures of Clinical Recovery

As for all Signature Bank participants, in addition to the abovementioned i-iii measures of recovery (personal recovery and citizenship), participants enrolled in this study will also have completed these other iv-ix clinical measures (clinical recovery) through their ongoing participation in the Signature Bank data collection (T1, T2, T3, and T4). Table 2 provides a summary of the main characteristics of measures i–ix.

### 2.3. Statistical Analysis

After controlling for baseline sociodemographic and clinical characteristics, analyses will all be performed in R v3.3.0 [43]. We will use the psych package [44] for reliability analyses and the lavaan package [45] for Structural Equation Modelling (SEM). Ordinal or logistic regression will be used, and we will present odds ratios with 95% confidence intervals.

#### 2.3.1. Internal Consistency

To evaluate the internal consistency of the RSA, RAS, and CM, Cronbach alphas [46] will be estimated for each of the subscales.

#### 2.3.2. Convergent and Concurrent Validity

To better understand differences in subscales between diagnostic categories, we will perform analysis of variance (ANOVA) post hoc pairwise comparison tests, corrected with a Tukey test for multiple comparisons. Convergent/concurrent validity is a series of tests to see whether constructs that are expected to be related are, in fact, related [47].

## 3. Discussion

Self-reported questionnaires have indisputable benefits with respect to the different psychological dimensions experienced by patients and their experience of health services. Beside measures i–iii of personal recovery, citizenship, and measures iv-ix of clinical recovery, all Signature Bank participants, including participants for this specific study, also complete a series of other self-reported questionnaires (Appendix A). Nevertheless, patients are not always able to adequately self-report the various medications prescribed to them or to assess their own clinical progression. Moreover, a large number of people suffering from psychiatric disorders have other chronic disorders as well, and it may be difficult for some patients to recall all of the medical conditions with which they have been diagnosed during previous years. For the purpose of obtaining this key information, it is therefore appropriate to use additional methods that make it possible to validate information via multiple sources. The Signature Bank routinely obtains information relating to participants’ diagnosis and psychiatric medication from their attending psychiatrist. Meanwhile, a request is transmitted to the Commission d’Accès à l’Information du Québec and the Régie de l’Assurance Maladie du Québec each year for all participants who took part in the Signature Bank project during the previous 12 months. The following information will be requested for the preceding two-year period with the aim of fully understanding all the chronic diseases from which participants are suffering: past and present mental and physical illnesses, medications prescribed for past and present mental and physical illnesses, medical complications, and causes of death. This information request will be used to confirm the diagnoses obtained from psychiatrists while also contributing to making better diagnostic assessments based on all available medical information and to identifying physical disorders whose presence may be linked to mental disorders [48] that may hinder clinical recovery, civic participation, and personal recovery as well.

On the other hand, the Signature Bank collects human biological materials from the same participants. The list of requested human biological materials is detailed in Appendix B. It will thus be possible to conduct secondary analyses to further explore possible correlational links between all these measures and those of citizenship and personal recovery, but this is not an objective of this actual study. Indeed, we plan to assess the effects of a group intervention led by Peer Support Workers on future Signature Bank participants and with a control group. Inspired by programs of Therapeutic Patient Education, this intervention is a series of 10–90-min long co-learning workshops on civic recovery. Therapeutic Patient Education is a set of pedagogical techniques developed for the purpose of enabling health care professionals–here Peer Support Workers (PSWs)–to pass on their knowledge and expertise to (other) patients [49]. Therapeutic Patient Education is based on the idea that educating patients–not only treating them–to develop skills to better manage and adapt their lives to their condition contributes to personal health [50]. For instance, the efficacy of Therapeutic Patient Education has been shown for improving the oral health of patients living with psychotic disorders [51]. The principal purpose is to produce a therapeutic effect in addition to that of other interventions (pharmacotherapy, psychotherapy, physical therapy, etc.) It is designed to enable a patient or a group of patients (and families) to manage the treatment of their conditions and prevent avoidable complications while maintaining or improving health outcomes [52] and quality of life. With this preliminary study, it is also a matter of assessing the effects of this patient-provided intervention from PSWs not only with patient-reported outcome measures but also, and to our knowledge for the first time, with patient-developed outcome measures. PSWs will learn with participants via a series of co-learning workshops that they will organize and facilitate as focus group panels in a manner to simulate a typical peer support group. The difference of our experimental and transitional peer support groups to real community-based peer support groups is that (A) they will have to be facilitated by trained PSWs and (B) they will have a citizenship and personal recovery focus. They will also (C) have a fixed, predetermined duration (a series of 10 weekly 90-min workshops), and this is why they are said to be transitional. Indeed, as defined by the World Health Organization: “Peer support groups bring together people who have similar concerns so they can explore solutions to overcome shared challenges and feel supported by others who have had similar experiences and who may better understand each other’s situation. Peer support groups may be considered by group members as alternatives to, or complementary to, traditional mental health services. They are run by members for members, so the priorities are directly based on their needs and preferences. Peer support groups should ideally be independent from mental health and social services, although some services may facilitate and encourage the creation of peer support groups” [53]. As recovery-oriented mental health care and services are particularly recognizable by the presence of PSWs, who can in some way personify personalized mental health care and services, that is to say, centered on the person’s recovery project and not only on their symptoms, data from this research project will lay the ground for the evaluation of the effect of the intervention of PSWs on both clinical recovery and to the personal-civic recovery composite index and possibly on other dimensions.

Ethical considerations: Declaration of Helsinki protocols are being followed and patients will give written informed consent. The study was approved by the Research Ethics Committees of the Montreal Mental Health University Institute under number 2020-1948. For all participants of the Signature Bank, including those participating in the research presented in this manuscript, an overseeing mental health expert has ruled that all adult patients were deemed ethically and medically capable of consenting for their participation.

## Figures and Tables

**Table 1 jpm-10-00163-t001:** Correlations among dimensions–citizenship and recovery measures.

Dimension	1	2	3	4	5	6	7	8	9	10
1. CM-Self-determination	*0.67*									
2. CM-Respect by others	0.37	*0.74*								
3. CM-Involvement in community	0.18	0.30	*0.65*							
4. CM-Fundamental needs	0.36	0.46	0.32	*0.60*						
5. CM-Access to services	0.38	0.29	0.26	0.28	*0.60*					
6. RAS-Personal confidences	0.33	0.46	0.36	0.48	0.27	*0.86*				
7. RAS-Willingness to ask for help	0.34	0.45	0.33	0.43	0.38	0.75	*0.61*			
8. RAS-Goal and success orientation	0.48	0.44	0.30	0.52	0.33	0.77	0.67	*0.80*		
9. RAS-Reliance on others	0.27	0.40	0.39	0.30	0.22	0.51	0.46	0.49	*0.60*	
10. RAS-No domination by symptoms	0.37	0.21	0.37	0.23	0.37	0.55	0.47	0.58	0.41	*0.77*

Note: N = 174. Cronbach’s alpha in italic along the diagonal. All correction coefficients are *p* < 0.01. RAS = Recovery Assessment Scale; CM = Citizenship Measure.

**Table 2 jpm-10-00163-t002:** Summary of the main characteristics of measures i–ix.

Dimension	Name of the Instrument	Abbreviation	Number of Items	Reference
i—Personal recovery	Recovery Assessment Scale	RAS	24	[15]
ii—Citizenship	Citizenship Measure	CM	23	[16]
iii—Organizational recovery	Recovery Self-Assessment	RSA	32	[31]
iv—Anxiety	Anxiety State-Trait Anxiety Inventory Form Y6	STAI-Y6	6	[37]
v—Depression	Depression Patient Health Questionnaire	PHQ-9	9	[38]
vi—Alcohol Dependence	Alcohol Use Disorders Identification Test	AUDIT-10	10	[39]
vii—Drug Dependence	Drug Abuse Screening Test	DAST-10	10	[40]
viii—Psychosis	Psychosis Screening Questionnaire	PSQ	12	[41]
ix—Social functioning	World Health Organization Disability Assessment Schedule	WHODAS 2.0	12	[42]

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
