# Peer review of "Convergent and Concurrent Validity between Clinical Recovery and Personal-Civic Recovery in Mental Healthâ€"

_jpm, 2020, doi:10.3390/jpm10040163_

Round 1
Reviewer 1 Report
The paper thoroughly describes the problem and gives a detailed study protocol that covers the information needed to answer the proposed research question.
A minor comment is that it will be helpful for the readers if the paper includes definition of some of the terms used, such as CM showing convergent validity with RAS. Also, in the beginning of the Discussion section, the authors describe that their patient cohort will complete other self-reported questionnaires and biological tests and these data can be used for secondary analyses. This part might be more appropriate for the Methods section.
Author Response
Response to Reviewer 1 Comments
Point 1: The paper thoroughly describes the problem and gives a detailed study protocol that covers the information needed to answer the proposed research question.
Response 1: We want to thank the reviewer for this revision. Thank you for this comment.
Point 2: A minor comment is that it will be helpful for the readers if the paper includes definition of some of the terms used, such as CM showing convergent validity with RAS.
Response 2: Convergent and discriminant validity are both subtypes of construct validity. Given that the CM and the RAS are measures of constructs that theoretically should be related, convergent validity has been tested and already confirmed between the CM and the RAS by estimating correlation coefficients. We now plan to estimate correlation coefficients between the CM and RAS on one side (total of 47 items for this personal-civic recovery composite index), compared with six measures of clinical recovery on the other side (total of 59 items). This information has been added in section 2.2-Measurements.
Point 3: Also, in the beginning of the Discussion section, the authors describe that their patient cohort will complete other self-reported questionnaires and biological tests and these data can be used for secondary analyses. This part might be more appropriate for the Methods section.
Response 3: We have removed this sentence from the Discussion section: “We will thus be able to perform secondary analyzes to see if personal-civic recovery also correlates or not with these other measures.” And since this is not an objective of this actual project, we preferred not to introduce this reference to possible secondary analyzes in the Methods section, if this is OK.
Reviewer 2 Report
The study entitled “Convergent and Concurrent Validity between Clinical Recovery and Personal-Civic Recovery in Mental Health” proposes a protocol to examine the psychometric properties of several recovery-related instruments. Although I appreciate the importance of the study purpose, the present manuscript suffers several significant flaws. Please see my comments below.
- The Introduction does not provide any clear aims or purposes for readers to know that this study will examine psychometric properties of which instruments.
- I cannot understand why the authors described the procedure in the 2.1.1. Sample selection section. To me, sample selection is talking about how the authors select the sample, not how the participants involved in the study. Also, the use of measures i to iii, or iv to ix are confusing here. Although the authors have referred reader to the following section, it is still hard for the readers to find these measures.
- Why do the authors want to exclude those with a learning disability?
- The term “Power calculation” is not precise here. The authors are actually proposing sample size estimation. Power can only be calculated after you conduct statistical analyses.
- The sample size calculation (or called as power calculation by the authors) is inappropriate from my perspective. Specifically, it is unclear how the sample size for factor analysis is done. There is no information regarding the simulation and it is unclear whether the factor analysis mentioned here is exploratory factor analysis or confirmatory factor analysis. The claim of the size of 200 for measurement invariance has no evidence or citations to support. The present study will assess concurrent validity; however, the sample size calculation does not mention how the authors determine the size on concurrent validity.
- A confusing part is that the authors have collected part of the data and reported the internal consistency and correlations for Recovery Assessment Scale and Citizenship Measure. However, they also want to examine the internal consistency and concurrent validity for the two instruments.
- A minor thing is that the authors said that Cronbach’s alpha is in italic along the diagonal for Table 1. However, the values do not read like italic.
- In Statistical Analysis section is also confusing. (1) The authors mentioned that they want to test measurement invariance under the section of Internal Consistency. Measurement invariance should be done after the structure is verified by the confirmatory factor analysis not the Cronbach’s alpha. However, the authors have roughly mentioned the procedure of measurement invariance under the section of Construct validity “The Satorra and Bentler scaled chi-square difference [48] will be used for model comparisons: 1) the configural model where the thresholds will be constrained equally across sexes or diagnoses, 2) the weak invariance model where the thresholds and factor loadings will be constrained to be equal across sexes or diagnoses, and 3) the strict invariance model where the thresholds, the loadings, and the variance will be equal across groups”. (2) The authors said that they will conduct ordinal or logistic regression; however, they provide no information regarding how they will conduct the regression models. (3) The authors stated that analysis of variance will be used to understand differences in subscales between diagnostic categories; however, they provided no plans on how to group the participants according to the diagnostic categories. (4) It is unclear why the authors mentioned the following sentence in the Statistical Analysis section. Convergent-concurrent validity is a series of tests to see whether constructs that are expected to be related are, in fact, related.
- The authors have collected data across four time points. However, they have not mentioned how they will use the data from different time points. This is a serious problem as this shows that the authors did not have a clear analytic plan in their mind. This also reflects that the authors did not have a clear idea of what the study purpose or aim is.
Author Response
Response to Reviewer 2 Comments
Point 1: The Introduction does not provide any clear aims or purposes for readers to know that this study will examine psychometric properties of which instruments.
First, we want to thank the Reviewer for this thorough revision.
Response 1: The main objective of this project is to explore possible correlations between clinical recovery, personal recovery, and citizenship by using validated patient-reported outcome measures.
Point 2: I cannot understand why the authors described the procedure in the 2.1.1. Sample selection section. To me, sample selection is talking about how the authors select the sample, not how the participants involved in the study. Also, the use of measures i to iii, or iv to ix are confusing here. Although the authors have referred reader to the following section, it is still hard for the readers to find these measures.
Response 2: Sub-section 2.1.1 is now titled “Sample selection and procedure”. We have also included the following table to help the reader to find more easily the measures at the end of section 2.2-Measures.
Table 2: Summary of the main characteristics of measures i-ix
Dimension |
Name of the instrument |
Abbreviation |
Number of items |
Reference |
i- Personal recovery |
Recovery Assessment Scale |
RAS |
24 |
32 |
ii- Citizenship |
Citizenship Measure |
CM |
23 |
33 |
iii- Organizational recovery |
Recovery Self-Assessment |
RSA |
32 |
34 |
iv- Anxiety |
Anxiety State-Trait Anxiety Inventory Form Y6 |
STAI-Y6 |
6 |
42 |
v- Depression |
Depression Patient Health Questionnaire |
PHQ-9 |
9 |
43 |
vi- Alcohol Dependence |
Alcohol Use Disorders Identification Test |
AUDIT-10 |
10 |
44 |
vii- Drug Dependence |
Drug Abuse Screening Test |
DAST-10 |
10 |
45 |
viii- Psychosis |
Psychosis Screening Questionnaire |
PSQ |
12 |
46 |
ix- Social functioning |
World Health Organization Disability Assessment Schedule |
WHODAS 2.0 |
12 |
47 |
Point 3: Why do the authors want to exclude those with a learning disability?
Response 3: This exclusion criterion has been removed.
Point 4: The term “Power calculation” is not precise here. The authors are actually proposing sample size estimation. Power can only be calculated after you conduct statistical analyses.
Response 4: Section 2.1.3 is now sub-titled “Sample size estimation” (formally “Power calculation”).
Point 5: The sample size calculation (or called as power calculation by the authors) is inappropriate from my perspective. Specifically, it is unclear how the sample size for factor analysis is done. There is no information regarding the simulation and it is unclear whether the factor analysis mentioned here is exploratory factor analysis or confirmatory factor analysis. The claim of the size of 200 for measurement invariance has no evidence or citations to support. The present study will assess concurrent validity; however, the sample size calculation does not mention how the authors determine the size on concurrent validity.
Response 5: Due to the covid-9 pandemic, this study has been suspended. We do not know yet if or when it will be possible to resume. At this point, given this situation, we will not be able to assess measurement invariance with the same constructs being measured across some specified groups (e.g.: sex, diagnosis). We will not do exploratory factor analyses either, only descriptive analyses of results to all instruments and concurrent validity analyses between measures of personal-civic recovery and of clinical recovery. The paragraph of whole section 2.1.3 (Sample size estimation) now reads like this:
A sample-size determination analysis was done in G*power v. 3.1.9.4. In the convergent analyses, correlation should be superior to 0.3 in absolute value. To detect an effect of this magnitude or greater, using a 5% type I error, we need at least 82 participants. Increasing the sample by10% to be conservative leads to a sample-size of 92 participants thus are sample-size of 95 patients is sufficient to perform the planned analyses [29,30].
-
- Faul, F., Erdfelder, E., Lang, A. G., & Buchner, A. (2007). G*Power 3: a flexible statistical power analysis program for the social, behavioral, and biomedical sciences. Behavior research methods, 39(2), 175–191. https://doi.org/10.3758/bf03193146
- Lenth, R. Some Practical Guidelines for Effective Sample Size Determination. American Statistician, 2001;55(3). 187–193. https://doi.org/10.2527/jas.2006-449
Point 6: A confusing part is that the authors have collected part of the data and reported the internal consistency and correlations for Recovery Assessment Scale and Citizenship Measure. However, they also want to examine the internal consistency and concurrent validity for the two instruments.
Response 6: This is true. A previous study enabled us to report on the internal consistency and correlations for the Recovery Assessment Scale and the Citizenship Measure. With this study, we aim to further assess concurrent validity for the two instruments compared to 6 clinical recovery instruments. If the analyses are conclusive, this finding would tend to confirm that clinical recovery and personal-civic recovery are two distinct constructs. This is important because for pedagogical reasons, when we present on recovery in mental health, we almost always have to explain that recovery in not a synonym for being cured and that it is not the absence of psychiatric symptoms. This would thus be supported by such findings.
Point 7: A minor thing is that the authors said that Cronbach’s alpha is in italic along the diagonal for Table 1. However, the values do not read like italic.
Response 7: The values of the Cronbach’s alpha are now in italic along the diagonal (Table 1).
Point 8: In Statistical Analysis section is also confusing. (1) The authors mentioned that they want to test measurement invariance under the section of Internal Consistency. Measurement invariance should be done after the structure is verified by the confirmatory factor analysis not the Cronbach’s alpha. However, the authors have roughly mentioned the procedure of measurement invariance under the section of Construct validity “The Satorra and Bentler scaled chi-square difference [48] will be used for model comparisons: 1) the configural model where the thresholds will be constrained equally across sexes or diagnoses, 2) the weak invariance model where the thresholds and factor loadings will be constrained to be equal across sexes or diagnoses, and 3) the strict invariance model where the thresholds, the loadings, and the variance will be equal across groups”. (2) The authors said that they will conduct ordinal or logistic regression; however, they provide no information regarding how they will conduct the regression models. (3) The authors stated that analysis of variance will be used to understand differences in subscales between diagnostic categories; however, they provided no plans on how to group the participants according to the diagnostic categories. (4) It is unclear why the authors mentioned the following sentence in the Statistical Analysis section. Convergent-concurrent validity is a series of tests to see whether constructs that are expected to be related are, in fact, related.
Response 8: We no longer plan, for now (see Response 5), to test measurement invariance. The whole section 2.3.2 (Construct validity) has been replaced by Convergent and concurrent validity. This sentence has thus been retrieved: “The Satorra and Bentler scaled chi-square difference [48] will be used for model comparisons: 1) the configural model where the thresholds will be constrained equally across sexes or diagnoses, 2) the weak invariance model where the thresholds and factor loadings will be constrained to be equal across sexes or diagnoses, and 3) the strict invariance model where the thresholds, the loadings, and the variance will be equal across groups”.
Also, as mentioned in section 2.1.1 (Sample selection and procedure) in this study we are using 2 of the categories of mental or behavioural disorder from the World Health Organization International Classification of Disease – ICD 10 (categories F00-F99): (1) Schizophrenia and psychotic disorders (F20-F29), and (2) Anxiety or mood disorders (F30-F49). The participants are thus grouped according to these diagnostic categories from the World Health Organization International Classification of Disease (ICD-10).
Point 9: The authors have collected data across four time points. However, they have not mentioned how they will use the data from different time points. This is a serious problem as this shows that the authors did not have a clear analytic plan in their mind. This also reflects that the authors did not have a clear idea of what the study purpose or aim is.
Response 9: For this study, data is being collected at only one time point. It is with another study, subsequent to this one, that we will be able to verify whether an intervention of Peer Support Workers can produce effects, and this would indeed be a different research design (pre-post). For now, it is only with Signature Bank participants that the iv-xi clinical recovery measures are taken at different time points. We use the last time point of completion. In addition to measure iv-ix, 95 of these participants from the Signature Bank will also completed measures i-iii once, which is sufficient to meet the objectives of this project designed to assess convergent / divergent validity between clinical recovery and personal-civic recovery (please see Response 5).
Reviewer 3 Report
Please review the following list of recommended corrections.
- I would suggest the use of another term, rather than “personal-civic”. Any paper which uses that term was found after a search in different databases. Neither was found in the book which is referenced after its definition at the manuscript (reference number 13). That term could be replaced by another more accurate term that the authors should consider. I recommend using “civic participation”, which is considered that directly influences important decisions in humans’ lives.
- The use of language and grammar needs to be reviewed. For example, the sentence from line 16 to 18 creates confusion. “Based on their life narratives, measurement tools have also been developed and validated through participatory research programs by persons living with mental health problems or illnesses to assess personal-civic recovery”. It can be understood as persons living with mental health problems developed and validated measurement tools.
- The following sentence can create confusion “participants diagnosed with (a) psychotic disorders or (b) anxiety” from line 20. Do participants present the mental disorder, or they have previously presented with a mental disorder but now they are healthy? This need to be explain cleared to the readers.
- There is a 6-line sentence in the abstract (from line 25 to line 30). Can that sentence be restructured?
- There is some information without references (line 50-54), (67-72).
- The authors should mention the main aim in the introduction in a clearer way.
- Study Design and Population needs to be briefly described, currently this is too long.
- I recommend creating a table which shows that information from line 219 to line 226. So that the table could be referenced along Materials and Methods.
- Will the participants previous and current treatment be monitored? It could present a future limitation from the study.
- In line 167, the sentence “we have found…” is not appropriated due to not all the authors of that paper being included in this manuscript to make this statement. It can be replaced by “it was concluded” or similar.
- The same abbreviation has been described multiple times. “Recovery Self-Assessment” (RSA) (lines 67, 122, 166).
- Does the project have a timeline that can show the schedule of the research?
- Make sure that your references are described as the journal requests. It is recommended to prepare the references with a bibliography software package, such as EndNote, Reference Manager or Zotero to avoid typing mistakes and duplicated references. For example, review reference number 15.
Author Response
Response to Reviewer 3 Comments
We first want to thank the Reviewer for this review.
Point 1: I would suggest the use of another term, rather than “personal-civic”. Any paper which uses that term was found after a search in different databases. Neither was found in the book which is referenced after its definition at the manuscript (reference number 13). That term could be replaced by another more accurate term that the authors should consider. I recommend using “civic participation”, which is considered that directly influences important decisions in humans’ lives.
Response 1: Thank you for this suggestion. We have better separated the two constructs of citizenship and personal recovery throughout the text and including in the summary. We have also added “civic participation” to the keywords.
Point 2: The use of language and grammar needs to be reviewed. For example, the sentence from line 16 to 18 creates confusion. “Based on their life narratives, measurement tools have also been developed and validated through participatory research programs by persons living with mental health problems or illnesses to assess personal-civic recovery”. It can be understood as persons living with mental health problems developed and validated measurement tools.
Response 2: Indeed, that is what they did, especially with the Citizenship Measure (CM). It is therefore a Patient Reported Outcome (PRO) to a measure developed by patients. We think this is a noteworthy nuance for this manuscript for a special issue on PROs and Self-Tracking for Personalized Medicine.
Point 3: The following sentence can create confusion “participants diagnosed with (a) psychotic disorders or (b) anxiety” from line 20. Do participants present the mental disorder, or they have previously presented with a mental disorder but now they are healthy? This need to be explain cleared to the readers.
Response 3: This sentence has been included in the abstract: “All study participants are currently being treated and monitored after having been diagnosed either with (a) psychotic disorders or (b) anxiety and mood disorders.”
Point 4: There is a 6-line sentence in the abstract (from line 25 to line 30). Can that sentence be restructured?
Response 4: Certainly, and this long sentence now reads in these three shorter sentences: “Recovery-oriented mental health care and services are particularly recognizable by the presence of Peer Support Workers, who are persons with lived experience of recovery. Upon training, they can personify personalized mental health care and services, that is to say services that are centered on the person's recovery project and not only on their symptoms. Data from our overall research strategy will lay the ground for the evaluation of the effects of the intervention of Peer Support Workers on clinical recovery, citizenship and personal recovery.”
Point 5: There is some information without references (line 50-54), (67-72).
Response 5: These additional references have been included:
- Mental Health Commission of Canada. 2009. Toward recovery & well-being: A framework for a mental health strategy for Canada. (line 53)
- Simpson A, Hannigan B, Coffey M, Jones A, Barlow S, Cohen R, et al. Cross-national comparative mixed-methods case study of recovery-focused mental health care planning and co-ordination: Collaborative Care Planning Project (COCAPP). Health Serv Deliv Res 2016;4(5). (line 72)
Point 6: The authors should mention the main aim in the introduction in a clearer way.
Response 6: This sentence has been added and the end of the Introduction: “The main aim of this study is thus to firstly explore convergent and concurrent validity between these constructs.”
Point 7: I recommend creating a table which shows that information from line 219 to line 226. So that the table could be referenced along Materials and Methods.
Response 7: This table has been added at the end of section 2.2-Measures.
Table 2: Summary of the main characteristics of measures i-ix
Dimension |
Name of the instrument |
Abbreviation |
Number of items |
Reference |
i- Personal recovery |
Recovery Assessment Scale |
RAS |
24 |
32 |
ii- Citizenship |
Citizenship Measure |
CM |
23 |
33 |
iii- Organizational recovery |
Recovery Self-Assessment |
RSA |
32 |
34 |
iv- Anxiety |
Anxiety State-Trait Anxiety Inventory Form Y6 |
STAI-Y6 |
6 |
42 |
v- Depression |
Depression Patient Health Questionnaire |
PHQ-9 |
9 |
43 |
vi- Alcohol Dependence |
Alcohol Use Disorders Identification Test |
AUDIT-10 |
10 |
44 |
vii- Drug Dependence |
Drug Abuse Screening Test |
DAST-10 |
10 |
45 |
viii- Psychosis |
Psychosis Screening Questionnaire |
PSQ |
12 |
46 |
ix- Social functioning |
World Health Organization Disability Assessment Schedule |
WHODAS 2.0 |
12 |
47 |
Point 8: Will the participants previous and current treatment be monitored? It could present a future limitation from the study.
Response 8: This sentence has been included in the abstract (circa line 20): “All study participants are currently being treated and monitored after having been diagnosed either with (a) psychotic disorders or (b) anxiety and mood disorders.”
Point 9: In line 167, the sentence “we have found…” is not appropriated due to not all the authors of that paper being included in this manuscript to make this statement. It can be replaced by “it was concluded” or similar.
Response 9: This has been corrected.
Point 10: The same abbreviation has been described multiple times. “Recovery Self-Assessment” (RSA) (lines 67, 122, 166).
Response 10: This has been corrected.
Point 11: Does the project have a timeline that can show the schedule of the research?
Response 11: This information has been added (section 2.2-Measurements): “In total, 95 participants were met between September 1st 2019 and March 1st 2020. They all completed the abovementioned questionnaires and quantitative analyzes are on the way. We expect to submit the main findings by the end of the year 2020.”
Point 12: Make sure that your references are described as the journal requests. It is recommended to prepare the references with a bibliography software package, such as EndNote, Reference Manager or Zotero to avoid typing mistakes and duplicated references. For example, review reference number 15.
Response 12: Reference number 15 (now #16) has been corrected, and 33 DOI have been included for the references.
Reviewer 4 Report
This is an interesting well-written study protocol about reported recovery in mental health assessed with different scales. There is a question/concern that came while reading it:
-It has been reported that physical and mental comorbidity may have an important weight in the information provided by patients but it is not controlled in the study in any way, the same occurs with peer support workers and their influence in the reported recovery, I would like to have a better understanding about this decision.
-line 254, check writing error 'we will be thus be'
Author Response
Response to Reviewer 4 Comments
We first would like to thank the Reviewer for this review.
Point 1: This is an interesting well-written study protocol about reported recovery in mental health assessed with different scales.
Response 1: Thank you for this comment.
Point 2: It has been reported that physical and mental comorbidity may have an important weight in the information provided by patients but it is not controlled in the study in any way, the same occurs with peer support workers and their influence in the reported recovery, I would like to have a better understanding about this decision.
Response 2: As mentioned in section 3. Discussion, the information on the physical condition of study participants are not provided by these patients, nor by Peer Support Workers. It can be provided by the Régie de l'Assurance Maladie du Québec (Quebec Health Insurance), upon request being made to the Commission d'Accès à l'Information du Québec (Quebec Access to Information Commission).
Point 3: Line 254, check writing error 'we will be thus be'
Response 3: Thank you, this error has been corrected.
Round 2
Reviewer 2 Report
The authors have well responded to all of my prior comments.
Reviewer 3 Report
Thank you for your new version and the cover letter. I consider that the manuscript has been significantly improved and now warrants publication in JPM.